# Effect of Saliva Collection Methods on the Detection of Periodontium-Related Genetic and Epigenetic Biomarkers—A Pilot Study

**DOI:** 10.3390/ijms20194729

**Published:** 2019-09-24

**Authors:** Pingping Han, Sašo Ivanovski

**Affiliations:** School of Dentistry, Faculty of Health and Behavioural Sciences, The University of Queensland, Brisbane, QLD 4006, Australia; p.han@uq.edu.au

**Keywords:** saliva collection methods, epigenetic factors, DNA methylation

## Abstract

Different collection methods may influence the ability to detect and quantify biomarker levels in saliva, particularly in the expression of DNA/RNA methylation regulators of several inflammations and tissue turnover markers. This pilot study recruited five participants and unstimulated saliva were collected by either spitting or drooling, and the relative preference for each method was evaluated using a visual analogue scale. Subsequently, total RNA, gDNA and proteins were isolated using the Trizol method. Thereafter, a systematic evaluation was carried out on the potential effects of different saliva collection methods on periodontium-associated genes, DNA/RNA epigenetic factors and periodontium-related DNA methylation levels. The quantity and quality of DNA and RNA were comparable from different collection methods. Periodontium-related genes, DNA/RNA methylation epigenetic factors and periodontium-associated DNA methylation could be detected in the saliva sample, with a similar expression for both methods. The methylation of tumour necrosis factor-alpha gene promoter from drooling method showed a significant positive correlation (TNF α, r = 0.9) with clinical parameter (bleeding on probing-BOP). In conclusion, the method of saliva collection has a minimal impact on detecting periodontium-related genetic and epigenetic regulators in saliva. The pilot data shows that TNF α methylation may be correlated with clinical parameters.

## 1. Introduction

Saliva is a biofluid that contains a large number of biomolecules, such as proteins, enzymes, exosomes and nuclear acids, and hence represents a “mirror” of the oral and systemic health. It contains both human and microbial sources, and so potentially provides insight into the relationship of the host with the environment and therapeutic drug monitoring [1,2,3,4,5]. Human saliva is unique which is the most easily accessible non-invasive body fluid. It allows rapid, cheap and repeated collections, is safe and painless and poses minimal threat to the collector of contracting infectious agents [2,6,7,8,9,10,11]. Currently, available research mainly focuses on salivary proteins (e.g., cytokines) as biomarkers of respiratory and other immunological disorders in the field of clinical diagnostics [8,11]. However, the drawbacks of using saliva are a) lower concentration of salivary analytes (100–1000 folds lower) than that in the blood; b) variability in concentration and composition in individuals; and c) saliva collection, process and storage methods may influence the composition and concentration [7,12,13,14,15]. Therefore, it is of great importance to standardise the saliva collection methods and develop an easy analytical method to detect low concentrations of biomarkers.

Saliva is a very complex system which includes both locally secreted components and resident micro-organisms, as well as systemic metabolites reflecting remote processes [2,13]. Of particular interest to periodontology is the contribution of gingival crevicular fluid components, which may reflect the periodontal status of an individual [16]. As such, saliva has been proposed as a key source for periodontal diagnostic biomarker discovery [2,17], although this field is still in its infancy, particularly in salivary DNA/RNA epigenetics.

For unstimulated saliva collection, spitting and drooling are two main saliva collection methods, which are easy to apply for the entire workflow from the clinic to the research laboratory, thus increasing the feasibility of developing standardised collection methods [1,7,13]. Various studies have shown that saliva collection methods (whole-mouth unstimulated spitting/drooling,) may influence the salivary composition and affect the final specific biomarker detection [18,19,20,21,22]. Furthermore, a recent study demonstrated that the bacterial genomic DNA (gDNA) quantity and quality are comparable between various saliva collection methods (spittingand drooling) and DNA isolation methods (e.g., commercial kit or Trizol method) [23]. However, it remains unclear whether collection methods (e.g., spitting and drooling) have a major impact on human periodontium-specific mRNA expression, DNA/RNA methylation epigenetic factors and periodontium-specific DNA methylation levels.

Reversible DNA (cytosine-5-carbon, m5C) and RNA (N6-methyladenosine, m6A) methylation are dynamic chemical epigenetic processes that play a crucial role in turning on (or off) gene expression [24,25]. They are a mechanism for transmitting cellular and environmental signals that influence gene expression and are responsible for maintaining and transferring cell-specific gene expression patterns to progeny cells [24,25,26,27]. Epigenetic factors can deposit, interpret and eliminate epigenetic information. They are divided into distinct functional groups: epigenetic ‘writers’ or enzymes that modify DNA and RNA; epigenetic ‘readers’ with specific protein domains that recognize DNA or RNA targets; and epigenetic ‘erasers’ that can delete existing signals to make room for new modifications [28,29,30]. It is unclear whether salivary circulating mRNA and epigenetic factors (writers, readers and erasers) are influenced by the saliva collection methods.

The purpose of this pilot study was to 1) develop a facile, cost-effective method to extract RNA, gDNA and protein from a small amount of saliva sample (200 μL) using the Trizol method; 2) explore the effect of the saliva collection methods (spitting and drooling) on periodontium-related gene profile; 3) evaluate the influence of different saliva collection methods on DNA/RNA methylation regulator profile and periodontium-specific DNA gene promoter methylation levels.

## 2. Results

### 2.1. Periodontal Status of Participants

As shown in Table 1, the participants came from a diverse ethnic background, mixed-gender (3 males, 2 females) and were relatively young (age range: 24–38). No participant had periodontitis, as evidenced by a lack of pocketing of 4 mm or higher. BOP ranged from 5–41%, with 3 participants being less than 15% (defined as healthy according to the latest classification of gingival diseases), while 2 were over 15% (defined as gingivitis) [31].

### 2.2. Salivary Flow Rate, DNA/RNA/Protein Quantity and Quality

There was no significant difference detected in terms of salivary flow rate, total protein quantity and salivary densitybetween spitting anddrooling) (Figure 1a–c).All five participants preferred the spitting method in terms of comfort, as determined by the visual analogue scale (VAS) scores to the question ‘How comfortable was the saliva collection process’ (Table 2). There was no significant difference in the quantity of extracted protein, genomic DNA (gDNA) or RNA in saliva samples collected from the spitting and drooling methods (Figure 1d–g).

gDNA was used as a template for qPCR assays to determine the existence of bacteria (16s rRNA) and human (β-globin) material. Of note, the mean CT values of β-globin and 16s rRNA from both methods were comparable (*p* = 0.06) after a paired non-parametric Wilcoxon T-test (Figure 1h,i), indicating that collection methods (spitting and drooling) do not alter the mixture of both bacterial and human gDNA in saliva samples.

These findings indicate that the Trizol method is a feasible, cheap and easy method to obtain sufficient biological materials (gDNA, RNA, and protein) from a single sample for subsequent downstream analysis.

### 2.3. Osteogenic and Wnt Pathway-Related Gene Expression

Since saliva is a complex mix of biomolecules from different cell types and signalling pathway, we next examined whether the collection method would influence periodontium-related osteogenic markers, Wnt pathway regulators and long-non-coding RNAs (lncRNA). The RT-qPCR results showed that there was no difference between osteogenic (*ALP*, *COLI*, *OPN*, *OCN* and *OSX*) (Figure 2), Wnt pathway (*WNT3A*, *LRP5*, *AXIN2* and *CTNNB*) (Figure 3), and intergenic lncRNA (*linc00907* and *RP11-575B7.3*) (Figure 3), indicating that the saliva collection methods had no major effect on periodontium-associated gene expression.

### 2.4. DNA and RNA Methylation Epigenetic Factors Changes

The expression of epigenetic factors for DNA and RNA methylation was carried out using RT-PCR. mRNA expression of DNA methyltransferases (*DNMT1*, *3A* and *3B*) showed no significant difference between spitting and drooling (Figure 4a). The same was observed for methyl-binding protein (MBD1-4, MECP2) (Figure 4b) and CpG demethylases (TET 1, 2, 3) (Figure 4c); suggesting that the collection method does not alter the detection of DNA methylation epigenetic factors.

The epigenetic factors for m6A methylation were also examined. There were no significant difference found for m6A methyltransferase (*METTL3*, *14*, *WTAP*, *KIAA1429*) (Figure 5a), m6A-binding protein (YTHDF 1, 2) (Figure 5b) and demethylases (*FTO*, *ALKBH 5*) (Figure 5c); indicating that the detection of RNA methylation epigenetic regulators was not affected by spitting or drooling collection.

### 2.5. Four Periodontium-Associated DNA Methylation Levels in Saliva from Different Collection Methods

To ensure the stability of bisulfite-converted DNA (which resembles RNA), all the quantitative methylation-specific PCR (qMSP) experiments used in this paper were generated within 1 week according to the manufacturer’s instructions. MSP primers for the individual periodontium-associated gene was investigated using non-converted DNA and we were unable to detect any qPCR amplification, suggesting that the MSP primers are specific.

MSP primers were designed around CpG islands of *RUNX2* gene (a key periodontium marker) promoter region for methylated (M) and unmethylated (U) primer pairs (Figure 6a). qMSP was performed for *RUNX2* with only 200 pg of converted DNA template and it showed that there was no significant difference between spitting and drooling (Figure 6b) and the MSP primer specificity for *RUNX2* was further confirmed with gel electrophoresis with a 250 bp band (Figure 6c). The same principle of primer design and qMSP was applied to *CEMP1*, *IL 6* and *TNF α* gene promoter. In addition, there was no significant difference observed for *CEMP1*, *IL6* and *TNF α* gene promoter methylation level in saliva from the different collection methods (Figure 6d–f). These findings indicated that the detection of methylation levels could be achieved from as little as 200 pg of bisulfite-converted DNA as a template for the periodontium-specific genes. Further, the saliva collection methods had minimal effect on periodontium-specific DNA methylation markers.

Next, we examined whether genes DNA methylation levels are correlated with mRNA expression levels. There was no positive correlation for *RUNX2*, *CEMP1*, *IL 6* and *TNF α* between DNA methylation level and mRNA expressions (Figure 6g–j).

### 2.6. Correlation between Gene Methylation Levels and Clinical Parameter

Periodontium-associated genes promoters (*IL 6*, *TNF α*, *CEMP 1* and *RUNX 2*) DNA methylation level was correlated with clinical parameters (BOP) using a Spearman’s rank correlation test. There was a positive correlation between *TNF α* DNA methylationand BOP for both collection methods: spitting (Figure 7a; r = 0.15, *p* = 0.83) and drooling (Figure 7b; r = 0.9, *p* = 0.01), indicating that TNF α gene promoter methylation might be a biomarker for periodontal health. *CEMP 1*, *IL 6* and *RUNX 2* gene promoters no positive correlation with BOP.

## 3. Discussion

There is evidence demonstrating that whole-mouth unstimulated saliva collection methods (spitting and drooling) may alter the saliva composition and influence the quantification of salivary biomolecules [18,19,20,21,22,23]. While a recent study demonstrated bacterial genomic DNA was minimally influenced by saliva collection methods (spitting and drooling) [23], there is a paucity of data and knowledge with respect to the influence of saliva collection methods on human periodontium-associated mRNA expression, epigenetic factors and DNA methylation levels. This pilot study developed a facile, cheap and easy-to-use method to isolate gDNA, RNA and protein from a single saliva sample, and further demonstrated that saliva collection methods do not alter periodontium-associated mRNA expression, DNA methylation levels and epigenetic factors.

The composition of saliva may vary considerably according to the flow rate, the type of stimulation and the time of day [1,32,33], while salivary flow rate varies among individuals [1]. In the current study, all of the five participants had normal salivary gland functions, with a saliva flow rate >0.1 mL/min [34]. The participants reported that the spitting method was their preferred method of saliva collection according to VSA data. Furthermore, we successfully utilised the Trizol method to isolate mRNA, gDNA and protein from a single sample and there was no significant difference found between the spitting and drooling methods without affecting the mRNA, gDNA and protein yield, which is in line with previous reports showing that gDNA quantity and quality is not altered between spitting and drooling [23]. It was found that mixture of both human and bacterial DNA was comparable by the collection methods, with no significant difference of both β-globin and 16s rRNA.

The periodontium consists of alveolar bone, periodontal ligament, cementum and gingiva, and tissue homeostasis, turnover and degradation are regulated by various genes and pathways (e.g., osteogenic genes and Wnt pathway) [35,36,37]. This study aimed to investigate whether saliva is a good source for detecting periodontium-associated gene expression. We demonstrated that bone-related genes (*ALP*, *COLI*, *BSP*, *OPN* and *OSX*), canonical Wnt-pathway related genes (*WNT 3A*, *LRP5*, *AXIN2* and *CTNNB*) and intergenic long non-coding RNA (*linc00907* and *RP11-575B7.3*) could be detected in the samples, and the expression levels were not altered by the two collection methods.

In mammalian cells, epigenetic changes, including DNA methylation and RNA m6A methylation, have been implicated in several critical biological roles, including cellular proliferation, differentiation, and development of multiple organisms [38,39]. Thus, it is of importance to investigate whether saliva collection methods would influence the epigenetic factors for DNA and RNA methylation. Our study showed that there was no significant difference in DNA methylation and m6A methylation enzymes profiles (“writers”, “readers” and “erasers”) between the different saliva collection methods, suggesting that key epigenetic factors are stable in saliva collected by the two methods used in this study. Furthermore, various studies have demonstrated that dysregulated methylation of periodontium-associated genes *CEMP 1*, *RUNX 2*, *IL 6* and *TNF α* is associated with systemic conditions such as obesity and osteoarthritis [40,41,42,43]; however, it remained unclear whether these methylation markers were detectable in saliva samples. This pilot study confirmed that the methylation levels of promoters for *CEMP 1*, *RUNX 2*, *IL 6* and *TNF α* genes were present in saliva, and were minimally affected by collection methods.

Recent research demonstrated that alteration in methylation levels of *IL 6*, *TNF α* and *IL 8* gene promoters have been reported in association with periodontitis in either gingival tissues or oral epithelial cells [44,45,46,47]. On the contrary, Asaad et al. demonstrated that inflammation-associated genes *LINE-1*, *COX-2*, *IFN*-gamma and *TNF α* in gingival tissues in healthy and periodontitis patients were not influenced by periodontal treatment [48]. However, it remains unclear whether salivary periodontium-associated genes DNA methylation is correlated with periodontal clinical parameter. Our results showed that *TNF α* from drooling method had a significant positive correlation with periodontal clinical parameter BOP (r = 0.9), indicating that *TNF α* DNA methylation might be a biomarker for periodontal health.

There are a few limitations in this pilot study: 1) the results of this pilot study need to be confirmed in a study with a larger population cohort; 2) the quality of total RNA and gDNA needs to be improved, possibly by adding DNase and RNase treatment (prior to spectrum measurement) respectively; 3) The PicoGreeen assay will be used for RNA samples to determine gDNA contamination and test for double-stranded gDNA; 4) housekeeping genes GAPDH and β-actin will be used to generate stable data for the future study; 5) 100% methylated DNA and non-methylated DNA should be used as a positive control and negative controls for future qMSP studies. Bisulfite sequencing or pyrosequencing are required for future studies.

Despite the limitations of this pilot study, the data from this research showed that saliva collection methods (spitting and drooling) had minimal effect on DNA/RNA epigenetic factors and periodontium-specific genes DNA methylation, indicating that saliva collection either by spitting or by drooling is feasible for future in-depth salivary DNA/RNA epigenetic research.

## 4. Materials and Methods

### 4.1. Study Participant and Saliva Sample Collection

This pilot study included five participants, aged 24–38 year (32.4 ± 2.7) from Oral Health Center, School of Dentistry, The University of Queensland, Australia. Periodontal charting was performed for each participant to determine their periodontal pocket depth (PPD) and bleeding on probing (BOP). This study was approved by the University of Queensland Human Ethics Research Committee (HREC No. 2018001225) and informed consent was obtained from all participants. All participants were non-smokers, with no underlying systemic diseases (details shown in Table 1).

Participants refrained from eating and drinking for at least one hour prior to saliva sample collection. Prior to sample collection, the participants were asked to rinse their mouth with ~10 mL of water to remove the food debris. Whole unstimulated saliva samples were collected from each participant using two different methods: spitting and drooling [22,23], with a 30-min interval on the same day between the two collections. Spitting samples were collected by asking participants to spit saliva directly into a 50 mL sterile falcon tube (Sarstedt, Mawson Lakes, Australia); drooling samples were collected 30 min later by asking participants to pool saliva in the mouth (2–5 min) before drooling into a sterile 50 mL falcon tube. Saliva samples were collected within a 2–3 min timeframe for a volume of 1–3 mL. Fresh saliva samples were processed for downstream analysis. The saliva flow rate was calculated as flow rate = saliva volume/collection time. The salivary density was calculated as total protein (μg/mL) × flow rate (mL/min) as a protein secretion rate (total protein output per unit of time).

Participants were asked to respond to the question ‘The saliva sample collection procedure was comfortable’ immediately after each saliva collection procedure, by marking a visual analogue scale (VAS) from 0 (disagree) to 9 (agree).

### 4.2. Salivary RNA Extraction Using a Trizol^™^ Method

Total RNA, gDNA and protein were isolated from 200 μL of whole unstimulated saliva using the Trizol^™^ reagent (Invitrogen, ThermoFisher Scientific Australia, Scoresby, Australia) following the manufacturer’s instructions. Total RNA was extracted as described previously [49], with a slight modification. Briefly, 0.8 mL of Trizol reagent was added into 200 μL of saliva, then the samples were vortex-mixed (~30 s) and then 200 μL of molecular biology grade 100% chloroform (VWR, Tingalpa, Australia) was added, and the tube was vigorously vortex-mixed (~1 min) and incubated for 15 min on ice. The samples were then centrifuged at 12,000× *g* for 15 min at 4 °C. The upper aqueous layer (containing RNA; ~500 μL) was transferred to a new Eppendorf tube; an equal volume at 500 μL of molecular biology grade 100% isopropanol (Sigma-Aldrich, Australia) was added, and the samples were incubated on ice for 10 min and at –20 °C for 1 h. The interphase (containing gDNA) and an organic phase (containing protein) were reserved for gDNA and protein isolation. The samples were then centrifuged at 12,000× *g* for 10 min and the pellet was washed with 1 mL of 70% molecular biology grade ethanol (Thermos Fisher Scientific, Scoresby, Australia) and centrifuged at 12,000× *g* for 5 min at 4 °C. The pellet was air-dried, resuspended in 12μL of RNase/DNase-free ultrapure water (Thermos Fisher Scientific, Scoresby, Australia), and stored at −80 °C. The quality and quantity of RNA were measured using a Tecan Infinite M200 Pro Spectrophotometer (TECAN, Melbourne, Australia).

### 4.3. Salivary gDNA Isolation Using Trizol

After total RNA isolation, the interphase and organic phase (~ 700 μL) were used for gDNA and protein isolation. For gDNA isolation, any remaining aqueous phase was removed to avoid RNA contamination. 300 μL of 100% molecular biology grade ethanol were added and the samples were mixed by inverting the tube several times. Following incubation for 3 min, the samples were centrifuged at 2000× *g* for 5 min at 4 °C to pellet the DNA. The phenol-ethanol supernatant was then transferred to a new tube for protein isolation. The DNA pellets were then resuspended in 1 mL of 0.1 M sodium citrate in 10% ethanol, pH 8.5, and incubated for 30 min, mixing occasionally by gentle inversion. The samples were centrifuged at 2000× *g* for 5 min at 4 °C; the pellet was then resuspended again into sodium citrate and incubated for another 30 min prior to a second centrifuge step at 2000× *g* for 5 min at 4 °C. Then the samples were washed with 2 mL of 75% molecular-grade ethanol and incubated for 20 min prior to centrifugation at 2000× *g* for 5 min at 4 °C. The supernatant was discarded and the DNA pellet was air-dried for 10 min. The pellet was resuspended in 200 μL of 8 mM sodium hydroxide (NaOH) and centrifuged at 12,000× *g* for 10 min at 4 °C to remove insoluble materials. The supernatant was then transferred to a new tube and the DNA yield was determined by a Tecan Infinite M200 Pro Spectrophotometer (TECAN, Melbourne, Australia).

### 4.4. Salivary Protein Isolation and Quantification

For protein isolation, the phenol-ethanol supernatant (~1 mL) was mixed with 1.5 mL of 100% molecular biology grade isopropanol and incubated at RT for 10 min. The samples were centrifuged at 12,000× *g* for 10 min at 4 °C to pellet the proteins and then resuspended in 2 mL of a wash buffer (0.3 M guanidine hydrochloride in 95% ethanol). The samples were incubated for 20 min and centrifuged at 7500× *g* for 5 min at 4 °C. The washing step was repeated twice. Then 2 mL of 100% molecular biology grade ethanol was added and vortex-mixed briefly (3 s) and incubated for 20 min. The samples were centrifuged at 7500× *g* for 5 min at 4 °C and the supernatant was discarded. The protein pellet was air-dried for 10 min and then resuspended in 200 μL of 1% sodium dodecyl sulphate (SDS) at 50 °C in a heat block. The samples were centrifuged at 10,000× *g* for 10 min at 4 °C to remove insoluble materials and then the supernatant was transferred to a new tube. The total protein was measured by a Pierce™ BCA Protein Assay Kit according to the manufacturer’s protocol (ThermoFisher Scientific, Scoresby, Australia). The absorbance was measured at 562 nm following a 30-min room-temperature incubation. The salivary protein, gDNA and RNA extractions were performed in triplicates.

### 4.5. gDNA Real-Time Quantitative PCR (qPCR)

gDNA qPCR was performed to determine the human and bacterial gDNA in saliva samples. Human β-globin gene (forward primer: 5′-CAACTTCATC CACGTTCACC-3′, reverse primer: 5′-GAAGAGCCAAGGACAGCTAC-3′) was used to amplify human gDNA, while bacterial 16S rRNA gene (forward primer: 5′-CGGCAACGAGCGCAACCC3′, reverse primer: 5′-CCATTGTAGCACGTGTAGCC-3′) was carried out to determine the bacterial gDNA. The reaction was comprised of 5 μL of 2 x PowerUp SYBR Green Master mix (ThermoFisher Scientific, Scoresby, Australia), 100 μM of forward and reverse primers and 10 ng of gDNA template. qPCR reaction (10 μL) was performed in StepOnePlus PCR equipment (Applied Biosystems, Australia) according to the manufacturer’s instructions. Each PCR reaction was performed in triplicates for each sample.

### 4.6. mRNA Expression for Osteogenic, Wnt-Related Genes and DNA/RNA Methylation Regulators

The effect of spitting and drooling on osteogenic marker expression was assessed by quantitative reverse transcription polymerase chain reaction (RT-qPCR) to measure the mRNA expression of alkaline phosphatase (ALP), osteopontin (OPN), bone sialoprotein (BSP), collagen I (COLI) and osterix (OSX). Total RNA was isolated using Trizol^TM^ Reagent according to the manufacturer’s instructions. Complementary DNA was synthesized using a First Strand cDNA Synthesis Kit (ThermoFisher Scientific, Scoresby, Australia) following the manufacturer’s instructions. RT-qPCR was performed using SYBR Green detection reagent in a StepOnePlus instrument (Applied Biosystems, Scoresby, Australia). The relative mRNA expressions were assayed and normalized against the housekeeping gene 18s rRNA. Each sample was performed in triplicate. The mean cycle threshold (Ct) value of each target gene was normalized against the Ct value of 18s and the relative expression calculated using the following formula: 2 ^−(normalized average Cts)^.

The same method was applied to the canonical Wnt pathway-related genes: Wingless-3A (*WNT3A*), low-density lipoprotein receptor-related protein 5 (*LRP5*), axis inhibition protein 2 (*AXIN2*), and beta-cadherin-associated protein (*CTNNB*); and the periodontium-associated intergenic long-noncoding RNAs (retrieved from FANTOM database): *linc00907* and *RP11-575B7.3*.

Furthermore, DNA methylation-related regulators were also assessed using RT-qPCR, including “writers”: DNA methyltransferase 1, 3A, 3B (*DNMT 1/3A/3B*); “readers”: methyl-binding protein (MBD) 1/2/3/4, methyl CpG binding protein 2 (*MECP2*); and “erasers”: ten-eleven translocation methylcytosine dioxygenase (TET) 1/2/3. RNA methylation-associated proteins were also measured, including “writers”: N6-adenosine-methyltransferase 70 kDa subunit 3/14 (*METTL 3/14*), Pre-mRNA-splicing regulator (WTAP), Vir Like M6A Methyltransferase Associated (*KIAA1429*); “readers”: YTH N6-Methyladenosine RNA Binding Protein 1/2/3 (YTHDF 1/2/3); and “erasers”: fat mass and obesity-associated protein (*FTO*), RNA demethylase alkB homolog 5 (*ALKBH 5*).

### 4.7. Bisulfite Conversion and Quantitative Methylation-Specific PCR (qMSP)

Isolated gDNA (500 ng - 1 μg) was bisulfite-converted with an EpiJET Bisulfite Conversion Kit (ThermoFisher Scientific, Scoresby, Australia). Modified DNA was resuspended in 15 μL of elution buffer and stored at − 20 °C. Purity and quantity of converted DNA (a single-stranded molecule which resembles RNA) were measured for RNA with a Tecan Infinite M200 Pro Spectrophotometer.

The qMSP primer pairs of periodontium-related genes (*CEMP1*), Runt-related transcription factor 2 (*RUNX2*) and inflammation-associated gene (interleukin 6 (*IL 6*), TNF α) were pre-designed around the CpG islands and the transcription start site (TSS) by an online tool MethPrimer (http://www.urogene.org/methprimer/) [50]. Both methylation and unmethylation primers for each gene were assessed using gDNA and were found not to amplify, indicating the specificity of primer pairs used in this study (Table 3).

qMSP was performed using a StepOnePlus PCR equipment (Applied Biosystems, Scoresby, Australia), consisting of 2 x PowerUp SYBR Green Master mix (ThermoFisher Scientific, Scoresby, Australia), 100 μM of forward and reverse primers and 200 pg of converted DNA template. Each PCR reaction was performed in triplicates for each sample. The methylated CTm value was normalized against unmethylated CTu value, where ΔCT = CTm − CTu. The relative methylation expression was calculated as 2^−ΔCT^. The relative methylated level (%) of total CpG islands was calculated using the following formula: $\frac{2-\Delta \mathrm{CT}}{2-\Delta \mathrm{CT}+1}$ × 100%. The specificity of qMSP was further confirmed by running 1 μL qPCR products on a 2% agarose gel.

### 4.8. Statistical Analysis

All the data are presented as mean ± standard mean of error (SEM) followed by a paired non-parametric T-test (Wilcoxon) analysis. A value of *p* < 0.05 was considered statically significant. Moreover, Spearman’s rank correlation test was used to determine the correlation between relative BOP, mRNA expression and methylation level.

## 5. Conclusions

In summary, total RNA, gDNA and protein can be successfully isolated from a single sample using the Trizol method; and the means of saliva sample collection (spitting and drooling) does not significantly influence the salivary periodontium-associated mRNA expression, epigenetic factor profiles and periodontium-related DNA methylation levels. Based on our findings, a saliva sample is a feasible source for epigenetic biomarker research, particularly DNA methylation. Our findings contribute to efforts to understand the influence of different saliva collection protocols on genetic and epigenetic biomarker expression for salivary diagnostics.

## Figures and Tables

**Figure 1 ijms-20-04729-f001:**
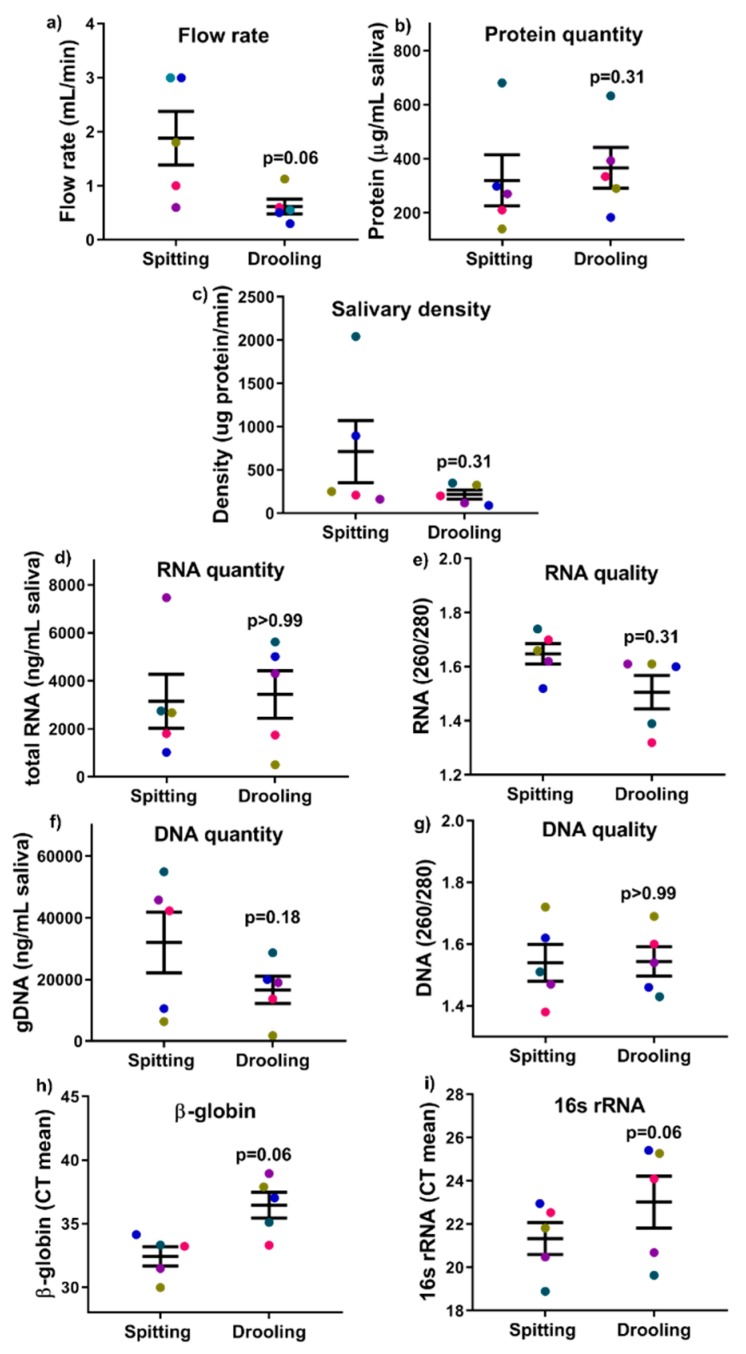
The effect of collection methods on the salivary flow rate (**a**), total protein quantity (**b**), salivary density (**c**), total RNA (**d**,**e**), DNA (**f**,**g**) quantity and quality, and human β-globin and bacterial 16s rRNA qPCR (**h**,**i**). Each coloured dot in Figure 1 (scatter graph) represents an individual participant.

**Figure 2 ijms-20-04729-f002:**
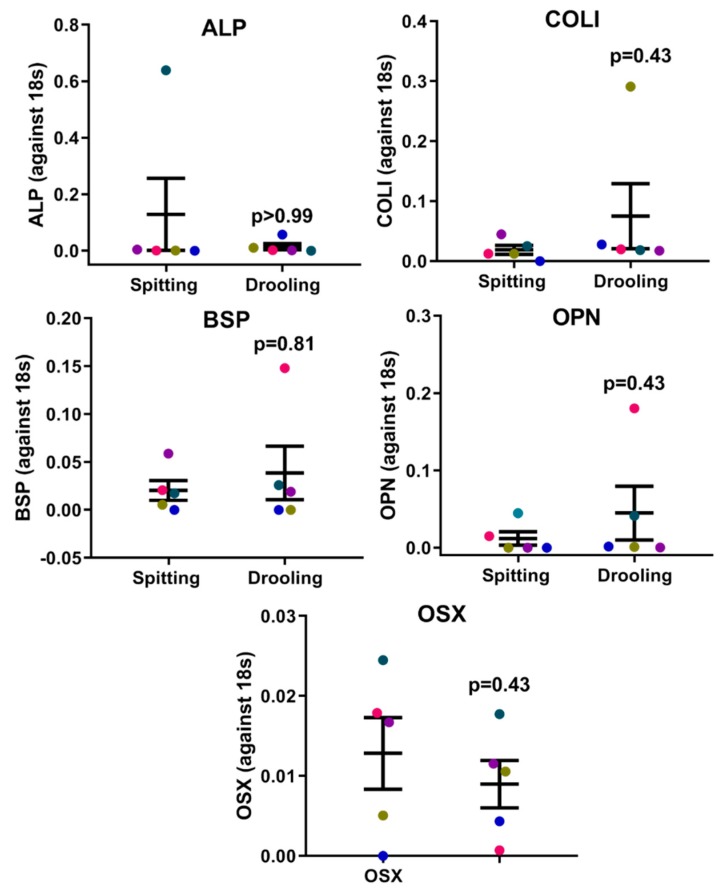
Osteogenic marker expression profiles on collection methods. There was no difference in *ALP*, *COLI*, *BSP*, *OPN* and *OSX* gene expression between spitting and drooling methods. Each coloured dot in Figure 2 (scatter graph) represents an individual participant.

**Figure 3 ijms-20-04729-f003:**
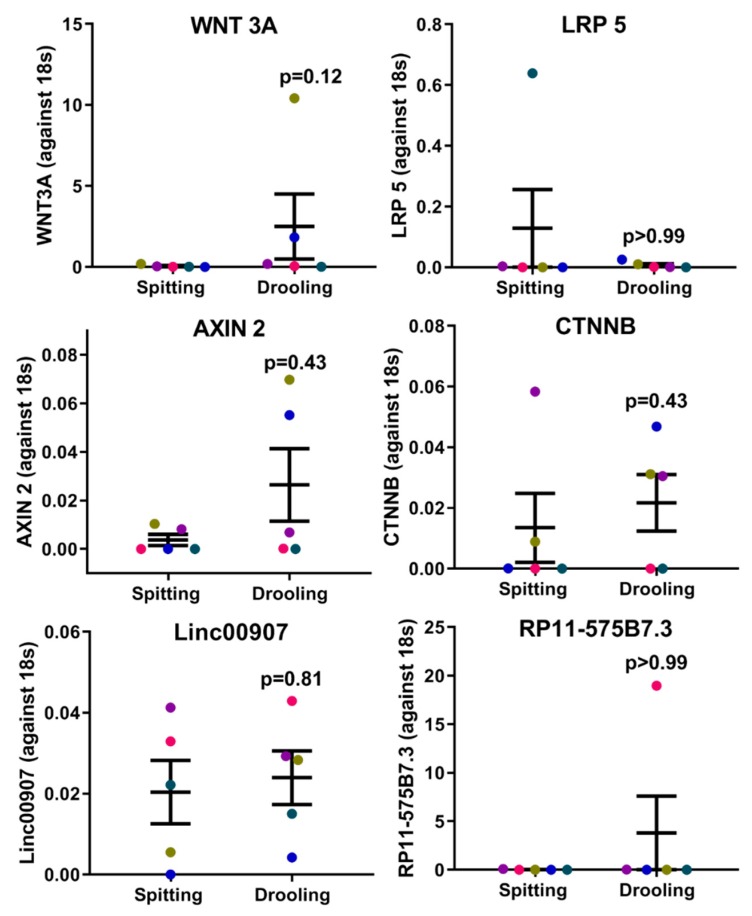
Wnt pathway-related regulators and periodontium-associated lncRNAs. There were no difference in the Wnt pathway (*WNT3A*, *LRP5*, *AXIN2*, and *CTNNB*) and periodontium-associated lncRNA (*linc00907* and *RP11-575B7.3*) gene expression between spitting and drooling methods. Each coloured dot in Figure 3 (scatter graph) represents an individual participant.

**Figure 4 ijms-20-04729-f004:**
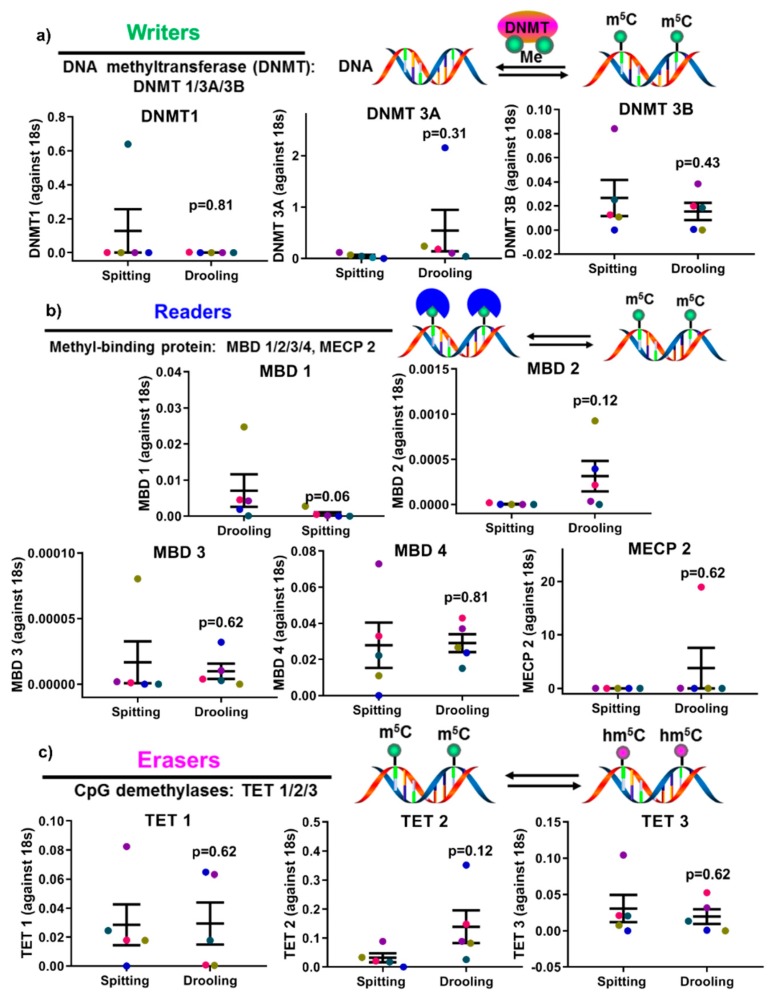
The influence of collection methods on salivary DNA methylation “writers” (**a**), “readers” (**b**) and “erasers” (**c**) profiles. There was no difference found “writers” (*DNMT1*, *3A*, and *3B*), “readers” (*MDB1-4*, *MECP2*) and “erasers” (*TET1-3*) from difference collection methods. Each coloured dot in Figure 4 (scatter graph) represents an individual participant.

**Figure 5 ijms-20-04729-f005:**
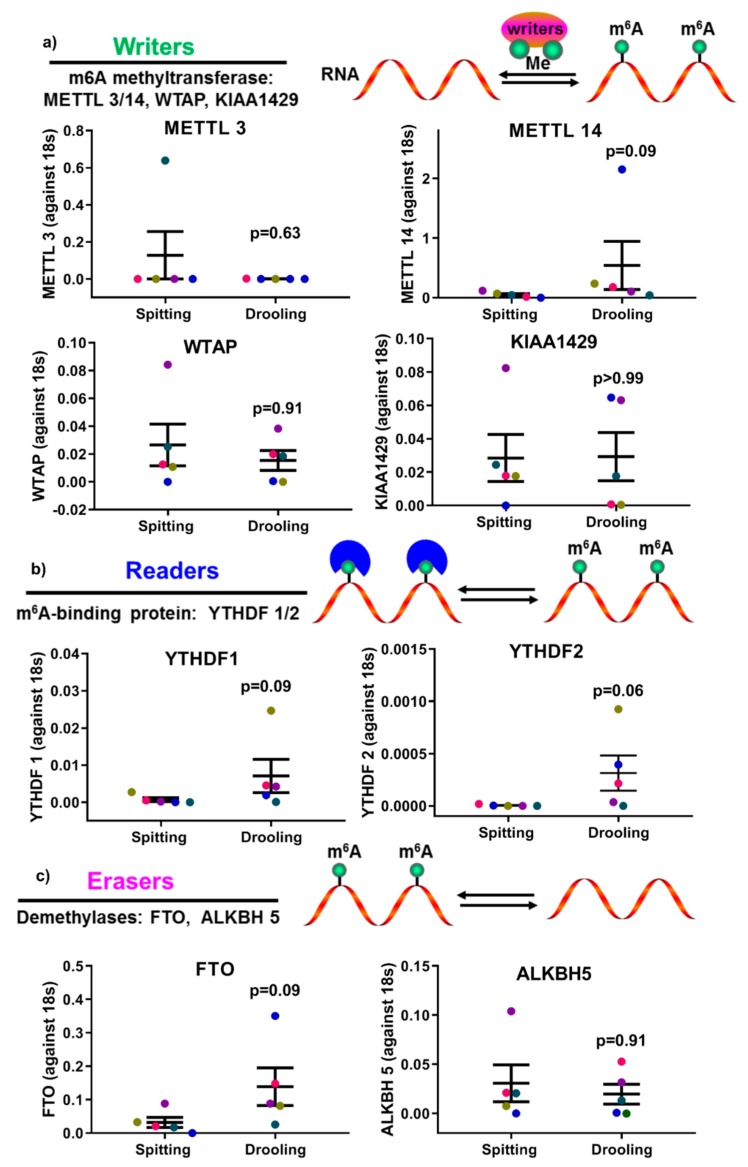
The impact of collection methods on salivary mRNA methylation (m6A) “writers” (**a**), “readers” (**b**) and “erasers” (**c**) profiles. There was no difference found “writers” (*METTL3/14*, *WTAP*, and *KIAA1429*), “readers” (*YTHDF 1*, *2*) and “erasers” (*FTO*, *ALKBH5*) from difference collection methods. Each coloured dot in Figure 5 (scatter graph) represents an individual participant.

**Figure 6 ijms-20-04729-f006:**
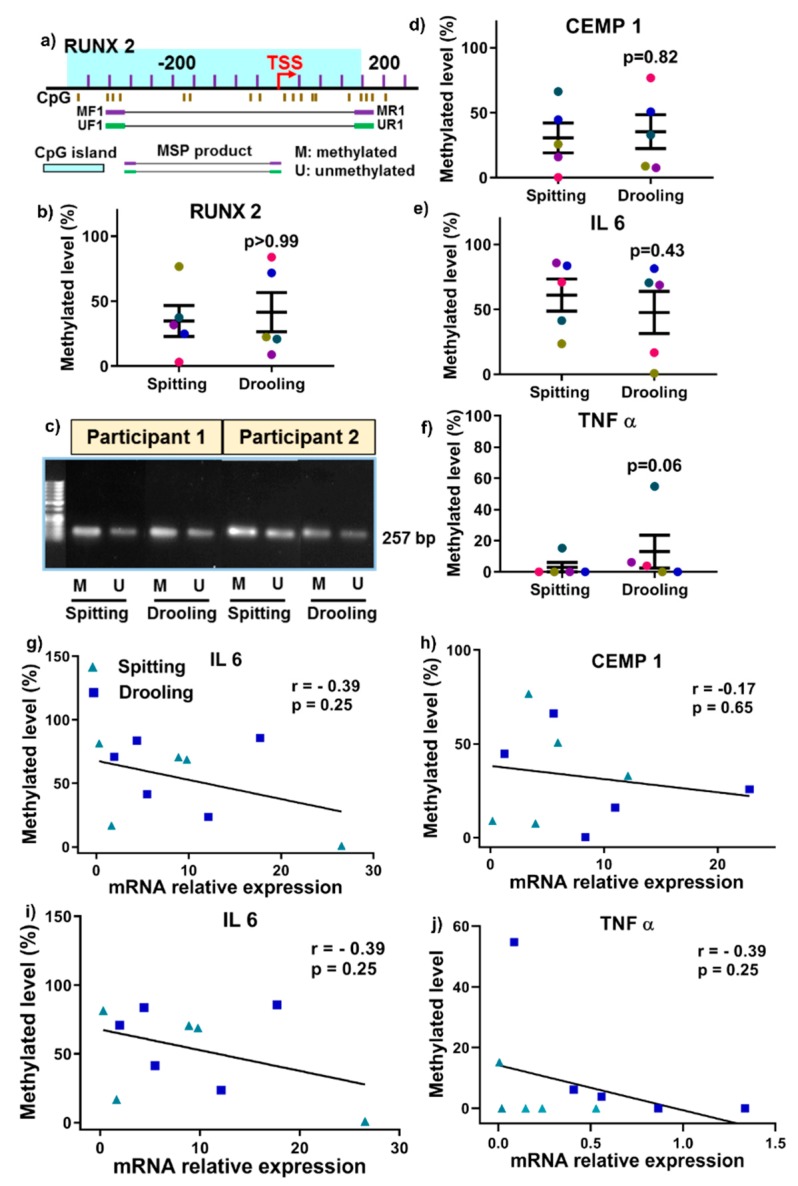
Methylation-specific PCR primer design (**a**), qMSP (**b**) and agarose gel (**c**) for RUNX2, a key osteogenic regulator. There was no significant change in CEMP1 (**d**), IL 6 (**e**) and TNF α (**f**) methylation from different collection methods. The relationship of DNA methylation level and mRNA expression was correlated using a Spearman Rank correlation test (**g**–**j**). Each coloured dot Each coloured dot in Figure 6 (scatter graph; **b**,**d**–**f**) represents an individual participant.

**Figure 7 ijms-20-04729-f007:**
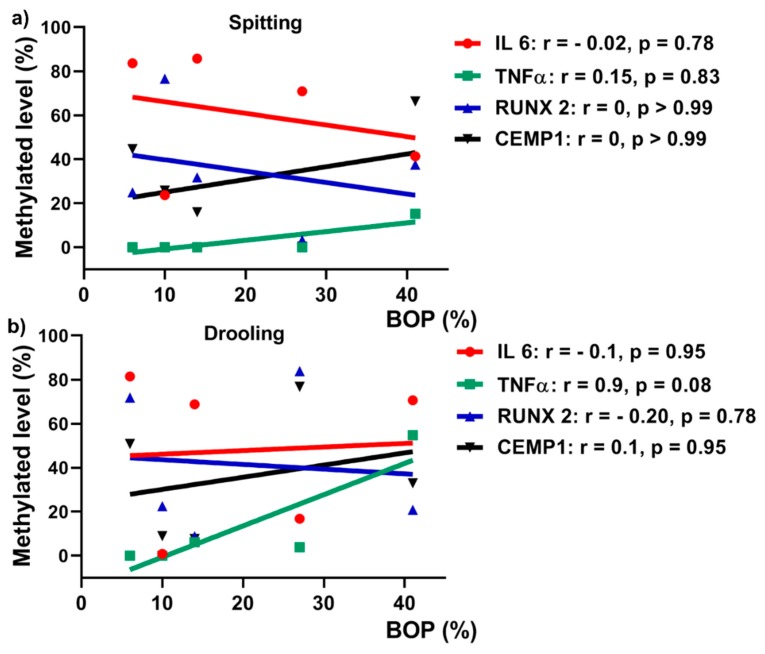
Pearson’s correlation between BOP and DNA methylation levels for spitting (**a**) and drooling (**b**) collection methods.

**Table 1 ijms-20-04729-t001:** Participant data.

	Gender	Age	PPD	BOP	Collection Method	Ethnicity
**1**	F	33	<4 mm	27%	Spitting, drooling	Asian
**2**	M	29	<4 mm	41%	Spitting, drooling	Asian
**3**	M	38	<4 mm	14%	Spitting, drooling	Caucasian
**4**	F	38	<4 mm	10%	Spitting, drooling	Asian
**5**	M	24	<4 mm	6%	Spitting, drooling	Caucasian

PPD: periodontal pocket depth; BOP: bleeding of probing.

**Table 2 ijms-20-04729-t002:** Data from VAS (scale 0–9: 0-disagree, 9-agree).

	Spitting	Drooling
**1**	9	4
**2**	9	2
**3**	9	3
**4**	9	1
**5**	9	1

**Table 3 ijms-20-04729-t003:** Methylated (M) and unmethylated (U) primer pairs used in this study for MSP-qPCR.

Gene		Forward (5′-3′)	Reverse (5′-3′)	Product (bp)
***RUNX2***	M pair	AGATTTCGTTCGGTAGTCGG	CTCACGTCGCTCATTTTACC	257
	U pair	AGATTTTGTTTGGTAGTTGG	CTCACATCACTCATTTTACC	257
***CEMP1***	M pair	TTACGAGGTGTAGAGGTTCGG	ACTCTCAAAACTAATAAAAATAACCCGT	200
	U pair	TTATGAGGTGTAGAGGTTTGGA	ACTCTCAAAACTAATAAAAATAACCCATTT	200
***IL 6***	M pair	GAGTTTATCGGGAACGAAAG	CTCCCTCACACAAAACTCGAC	133
	U pair	GAGTTTATTGGGAATGAAAG	CCCTCACACAAAACTCAACC	131
***TNF α***	M pair	GCGATGGAGAAGAAATCGAG	AACAACTACCTTTATATATCCCTAAAACG	153
	U pair	TGATGGAGAAGAAATTGAGAT	AACTACCTTTATATATCCCTAAAACAAAAA	149

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
