# Peer review of "Effect of Saliva Collection Methods on the Detection of Periodontium-Related Genetic and Epigenetic Biomarkers—A Pilot Study"

_ijms, 2019, doi:10.3390/ijms20194729_

Round 1

Reviewer 1 Report

Han and Ivanovski tested 2 collection methods for saliva and compared gDNA, RNA and protein using samples from 5 individuals. They did not observe large differences in the genes and factors analyzed. They conclude that the 2 methods tested have minimal impact on their measurements.  

- Authors state: “Whole unstimulated saliva samples were collected from each participant using two different methods: spitting and drooling [19, 20], with a week interval between the two collections. Spitting samples were collected by asking participants to spit saliva directly into a 50 mL sterile falcon tube (Sarstedt, Australia); drooling samples were collected 30 minutes later by asking participants to pool saliva in the mouth (2-5 minutes) before drooling into a sterile 50 mL falcon tube.” Is saliva sampled from each participant in the same day or 1 week later?

- A fluorescence-based method would be more accurate for determining concentration of total RNA and gDNA.

- mRNA should be changed to total RNA (line 270)

- gRNA should be changed to gDNA (line 271)

- Why is SDS, a denaturing agent, is added to resuspend proteins? SDS interferes with Bradford assay. BCA assay is more recommended for SDS-containing samples.

- Authors use 10 ng of gDNA template for qPCR, though measurement based on spectrometry is probably unreal.

- Include pb of the amplicons

- Only one housekeeping gene is used (18s rRNA). Did authors checked for other housekeeping genes? For future studies I recommend to use 2-3 genes to guarantee stability in expression.

- Authors say “Purity and quantity of converted DNA were measured with a Tecan Infinite Spectrophotometer.” How did they made this measurement? DNA after bisulfite is not double stranded and is not complementary. Did they analyzed as RNA?

- “The methylated CT value was normalized against unmethylated Ct value, and the relative methylated level (%) was calculated using the following formula: [(2−(normalized average Cts))/ (2−(normalized average Cts)+1)] x 100%.” Please explain how methylation was estimated. This they use 100% methylated control? Unmethylated control?

- Please explain the importance of the salivary flow rate in relation to extraction of DNA, RNA and protein.

- Is there an explanation for the difference in RNA quality between methods? Maybe this is due to small sample size.

- Perhaps each patient could be represented in the figures with a different color for a pair-wise comparison of the different collection methods. This can help with outliers. It seems that data is missing from a patient in some cases.

- Figure 6B, change to Methylated level.

- Line 146-147 please check. Each amplicon should have their own primers.

- Figure 7B is missing?

- Check spelling of pallet, salvia, demonstrsted, unlcear.

- Correlation between BOP and DNA methylation are presented using all data (5+5). Correlation should be evaluated separating methods, especially in the case of TNFα that was almost statistically different between methods. When separating methods, TNFα still shows positive correlation with BOP?

- Conclusion paragraph seems incomplete and does not say much.

Reviewer 2 Report

This is a well written article that supports that saliva can be used a feasible and reliable source for epigenetic research and DNA methylation analysis.

The article also shows that two saliva collection methods, namely spitting and drooling, do not have an impacr on mRNA expression analyses.

The major problem with this work has to do with the word "reliable". For sure, epigenetic analysis is feasible when nucleic acids and proteins are extracted from spit or drooled saliva via the Trizol method. The authors clearly show this feasibility. However, from there to say that these analyses are reliable, is a big step. For such a statement to be made, the authors should have compared the epigenetic DNA biomarkers they have analysed between the DNA extracted from saliva with their method and gDNA extracted from whole blood with a DNA dedicated extraction kit. Unless there is a reason to believe that gDNA methylation in saliva is different from gDNA methylation in blood.

Another methodological question has to do with the yields. The protein yield is reasonable and according to expactations, however the RNA yield seems tobe too high, actually ten times higher than what is usually expected from saliva. This raises the question of potential DNA contamination, especially since the authors have not performed DNase digestion. One way to address this point would be by performing spectrofluorimetric measurements by Picogreen in the RNA samples to rule out DNA contamination. Similar Picogreen measurement could be done in the DNA extracts to see what is the yield of double stranded DNA. For DNA extracts, RNase digestion has not been performed either.

Minor  remarks:

Line 94-96: Lower Ct for 16S rRNA target does not indicate higher abundance of bacterial gDNA. The 16SrRNA gene is present in multiple copies. To compare bacteriual and human DNA abundances, qPCR for bacterial and human would have to be done. Please delete this statement.

The authors use the woeding "methylation levels of IL6, TNFa and IL8". Thie wording should be "methylation levels of IL6, TNFa and IL8 coding genes".

In the Materials and methods, it is not clear if the spitting and drooling collection methods were applied a week apart or 30minutes apart.

Reviewer 3 Report

COMMENTS TO THE AUTHOR(S)

The manuscript is well written and well organized however it requires major revisions before to be accepted for publication in IJMS journal.

General comments:

The work presented by Han et al is very interesting and represent an hot topic in the field of saliva analysis. In fact, the main problem with saliva analysis is the impact of the collection procedures on the chemical composition of such fluid. This aspect limit the use of saliva analysis in the clinical field. However, the limited number of participants enrolled in this paper does not allow to obtain any statistically data regarding the impact of the collection procedures, then I suggest to increase the number of participants. The authors can perform a power analysis in order to estimate the right number. In addition, please include in the introduction the reasons of testing only spitting and drooling methods.

Specific comments:

L13, I think that more than one protein are included in the evaluation process, thus change “protein” with “proteins”.

L20, please move the “(TNF alpha, r = 0.72)” after “… a significant positive correlation…”

L28, saliva analysis is currently used in the field of therapeutic drug monitoring, so I suggest to include the following articles in order to improve the scientific background of the introduction.

10.1371/journal.pone.0028182

10.1016/j.microc.2017.04.033

10.1016/j.microc.2017.02.010

L38 and L47-49, I totally agree with the authors regarding the main drawbacks of saliva analysis. I suggest to include the following articles in order to improve the scientific background of the introduction.

10.1016/j.microc.2017.02.032

10.1371/journal.pone.0114430

L89-90, please modify the digit numbers according to the variability of the CT analysis. In addition, please include the type of test (e.g. t-test) used to compare groups and the resulting p number.

L93, as before, please include the p number

L97, looking the figure, seems that the authors missed the word “quantity” after protein. Please check.

Figure 1, 2, 3, 4, 5 and 6, please explain in the caption the kind of graph used by the authors.

L139, I suggest to explain the stability conditions tested.

L187, the term validation has a known meaning and did not fit well with the results obtained by the authors. In fact from an analytical point of view, a method is validated after performing several experiments in order to estimate quality parameters. I suggest to modify this word with one more appropriate.

L208, please check the spelling of the word ”salvia”.

L212, did the authors have an idea about of these results?

L234, please modify the digit number of the standard deviation.

L242, please include the liquid and the volume used to rinse the oral cavity before to collect saliva.

L244, please explain the reasons for collecting saliva samples with a time span of 1 week. I disagree with this experimental plan because is not possible to compare results.

L245, please explain the type of plastic material used. Did the authors tested any possible interaction between plastic and target analytes?

L249, please check the sentence and include that the density of saliva was considered unit.

L258-259, please include the vortex-mixing timing for both steps.

L260, please include the volume of aqueous layer used.

Materials and Methods, please include all the solvent purity.

L267-268, please include all the instrumental information used to evaluate the quality and quantity of RNA.

L286-287, please include the initial amount of phenol-ethanol supernatant and the incubation parameters.

L298, please include these data in the article.

L309, please include these data in the article.

L361, please check the sentence since is not clear the meaning of faster flow rate.

Round 2

Reviewer 2 Report

thank you for the revision and listing the limitations of the study.

Reviewer 3 Report

Dear Authors,

the present version of the paper is suitable for publication in IJMS journal.

Regards.